# Passivation and pH-Induced Precipitation during Anodic Polarization of Steel in Aluminate Electrolytes as a Precondition for Plasma Electrolytic Oxidation

Roy Morgenstern [1,*], Claudia Albero Rojas [1], Frank Simchen [1], Vanessa Meinhold [2], Thomas Mehner [1] and Thomas Lampke [1]

1   Materials and Surface Engineering Group, Faculty of Mechanical Engineering, Chemnitz University of Technology, 09107 Chemnitz, Germany
2   FMT Flexible Montagetechnik GmbH, 09212 Limbach-Oberfrohna, Germany
*   Correspondence: roy.morgenstern@mb.tu-chemnitz.de; Tel.: +49-371-531-32818

**Abstract:** Potentiodynamic and potentiostatic polarization tests in the potential range between open circuit potential (OCP) − 0.1 V and OCP + 4 V were carried out in aluminate–phosphate electrolytes with an aluminate concentration of 0.2 mol/L and varying phosphates contents between 0 and 0.1 mol/L. The pH was adjusted between 11.5 and 12.0 due to phosphate and optional KOH addition. A high-strength, dual-phase steel, which is relevant for lightweight construction, served as the substrate material. The layer microstructure was investigated by optical and scanning electron microscopy. Energy-dispersive X-ray spectroscopy and Raman spectroscopy were used for element and phase analyses. We found that iron hydroxides or oxides are initially formed independently of the electrolyte composition at low potentials. At around 1 V vs. standard hydrogen electrode (SHE), the current density suddenly increases as a result of oxygen evolution, which causes a significant reduction in the pH value. Precipitation leads to the formation of porous layers with thicknesses of 10 μm to 20 μm. In the case of a pure aluminate solution, the layer mainly consists of amorphous alumina. When adding phosphate to the electrolyte, the layer additionally contains the hydrous phosphate evansite. At the highest phosphate content in the electrolyte, the highest P content and the most pronounced crack network were observed.

**Keywords:** passivation; precipitation; polarization; aluminate; phosphate; pH; dual-phase steel

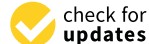



## 1. Introduction

Electrochemical passivation is considered an important prerequisite for plasma electrolytic oxidation (PEO) [1,2]. Passivation is generally understood as the deposition of a poorly soluble compound from the dissolved metal ions and ions of a corrosive solution after the solubility product has been exceeded [3]. This causes the formation of a protecting layer on the substrate, which kinetically inhibits both the anodic metal dissolution and the electrolysis of the water, i.e., the oxygen evolution at the anode. A significant part of the current occurs due to the outward migration of metal ions towards the passive layer/electrolyte interface and the migration of the oxygen ions in the opposite direction. Lohrengel summarizes the mechanisms of ion transport and passive layer growth according to the high-field model in [4]. With increasing oxide layer thickness, the anodic potential for maintaining the current must be continuously increased. As a result of the oxygen generation and the electrolyte evaporation due to Joule heating, a gas envelope forms on the anode surface. According to the model by Yerokhin, microarc initiation occurs above the breakdown potential between a quasi-cathode on the surface of the gas envelope and the anode [5].

In contrast to chemical elements such as Al, Nb, and Zr, Fe does not spontaneously form a dense and adherent oxide layer under humid conditions but a porous and loose

mixture of iron oxides and hydroxides, which is well known as rust [6]. However, numerous studies describe the anodic passivation of Fe in strongly alkaline solutions above a pH of 13 [7–11]. At low anodic polarization, iron(II) oxide and hydroxide are initially formed according to Equations (1) and (2) [7]. A corrosion-protective passivation is observed after further oxidation to iron(II,III) oxide (magnetite) according to Equations (3) and (4) [7–9]. With increasing anodic potential, iron(II,III) oxide is oxidized to iron(III) oxide or hydroxide [7–9], resulting in a multilayer structure with a higher proportion of Fe(II) oxide close to the substrate and Fe(III) oxide and hydroxide on the surface [10]. The thickness of the passive layers is in the range of a few nanometers [11]. The corrosion and passivation behavior of high-strength multiphase steels, e.g., dual phase (DP) steels, which consist of the ferrite and martensite phases [12], are more complex than in case of single-phase ferritic iron. It is known from numerous corrosion studies that at pH values around 7, martensite behaves more electrochemically noble and that the ferrite phase corrodes preferentially. Therefore, the corrosion rate increases with increasing martensite content on the surface [13–16]. However, in alkaline solutions, the anodic passivation of the ferrite phase due to Equations (1) to (6) might be supported by the galvanic coupling with martensite. Abdo et al. found that a more stable passivation layer was formed during anodic polarization in 0.8 mol/L NaOH after a DP heat treatment compared with the normalized ferrite-perlite condition [17].

$$Fe + 3H_2O \rightarrow FeO + 2H_3O^+ + 2e^- \tag{1}$$

$$Fe + 4H_2O \rightarrow Fe(OH)_2 + 2H_3O^+ + 2e^- \tag{2}$$

$$3Fe(OH)_2 + 2OH^- \rightarrow Fe_3O_4 + 4H_2O + 2e^- \tag{3}$$

$$3FeO + 2OH^- \rightarrow Fe_3O_4 + H_2O + 2e^- \tag{4}$$

$$2Fe_3O_4 + 2OH^- \rightarrow 3Fe_2O_3 + H_2O + 2e^- \tag{5}$$

$$Fe_3O_4 + OH^- + H_2O \rightarrow 3FeO(OH) + e^- \tag{6}$$

The formation of a corrosion-protecting layer can be promoted by the addition of further anions to the solution. For example, the addition of silicate leads to the formation of a thin protective layer, effectively inhibiting Fe dissolution at pH 12 [18]. Pronounced passivation at pH 12 was also observed in electrolytes containing aluminate [19]. Two different mechanisms for the formation of an aluminum oxide or aluminum hydroxide surface layer from aluminate-containing electrolytes have been proposed in the literature: electrochemical oxidation and the precipitation reaction [20,21]. Electrochemical oxidation of aluminate can produce insoluble aluminum hydroxide according to Equation (7) or alumina according to Equation (8). Due to oxygen evolution at the anode, a drop in the pH value is expected. With a decreasing pH value and thus a decreasing ratio of $OH^-$ to $Al^{3+}$, polymers of the type $[Al(OH)_4]_n (OH)_2^{(n+2)-}$ are initially formed. In the pH range between 8.2 and 9.3 and at an $OH^-/Al^{3+}$ ratio in the range of 3.01 to 3.3, colloidal $Al(OH)_3$ eventually precipitates [21,22]. According to Ginsberg et al., crystalline $Al(OH)_3$ precipitates in the pH range between 10 and 12.5 after prolonged storage [23]. In order to prevent premature precipitation, aluminate solutions in this pH range must be stabilized with complexing agents [20,24]. Furthermore, gels form between pH 8 and 10, with an increasing proportion of crystalline $AlO(OH)$ at increasing pH. Stable solutions of $Al^{3+}$ and $[Al(OH)_4]^-$ exist below pH 8 and above pH 13 [23]. For the precipitation of $Al(OH)_3$ as a result of anodic acidification, the pH value of the solution and the anodic potential must be set within narrow limits [24]. The precipitation reactions of $Al(OH)_3$ and $Al_2O_3$ are represented in simplified form in Equations (9) and (10) [20,21].

$$[Al(OH)_4]^- \rightarrow Al(OH)_3 + 1/2 O_2 + 2H_2O + e^- \tag{7}$$

$$2[Al(OH)_4]^- \rightarrow Al_2O_3 + 1/2 O_2 + 4H_2O + 2e^- \tag{8}$$

$$[Al(OH)_4]^- + H^+ \rightarrow Al(OH)_3 + H_2O \tag{9}$$

$$2[\text{Al(OH)}_4]^- + 2\text{H}^+ \rightarrow \text{Al}_2\text{O}_3 + 5\text{H}_2\text{O} \tag{10}$$

At high anodic potentials of several 100 V and with intense oxygen evolution, Karpushenkov et al. concluded that the layer formation is mainly based on a precipitation reaction according to Equations (9) or (10) [21]. For alkaline electrolytes, which additionally contain hydrogen phosphate ions, Li et al. proposed a precipitation reaction at the anode, which leads to the deposition of a mix of aluminum oxide and aluminum phosphate [25]. Equation (11) describes the overall reaction. In contrast, Kurze considered that a precipitation reaction caused by anodic acidification is unlikely, since the concentration of $\text{H}^+$ ions in alkaline media is very low. Furthermore, $\text{H}^+$ ions would be immediately repelled from the anode due to their positive charge, and a strong oxygen evolution would prevent the formation of an adherent oxide layer. They advocate the mechanism of electrochemical oxidation of aluminate ions according to Equations (7) or (8) [20].

$$3[\text{Al(OH)}_4]^- + [\text{HPO}_4]^{2-} \rightarrow \text{Al}_2\text{O}_3 \cdot \text{AlPO}_4 + 5\text{OH}^- + 4\text{H}_2\text{O} \tag{11}$$

Based on a broad design of experiments, Kurze described the production of dense and adherent layers by anodic polarization in the potential range up to 75 V, which mainly consist of amorphous, water-containing $\text{Al(OH)}_3$ [20]. As a result of the dehydration of the layer by drying in air or accelerated drying at elevated temperatures or under vacuum conditions, there is a significant reduction in volume, which leads to the formation of a crack network [20]. Li et al. described the formation of a surface layer, which mainly consists of Al, O, and P, at around 450 V (still below the ignition voltage), with the alumina aluminum phosphate (see Equation (11)) or aluminum phosphate phases predominating [18]. After the formation of Al-oxide- or Al-hydroxide-rich top layers and after the breakdown potential is exceeded, microarc discharges were initiated and thus the PEO process began on Fe substrates, similar to the PEO of Al alloys [20,21,25].

A temperature of about 7000 K to 10,000 K is reached in the center of the discharge channel [26]. There, substrate regions close to the surface, the passive layer, and anions from the electrolyte present at the quasi-cathode are vaporized and form plasma. This is followed by a region where the oxide formation reaction mainly takes place, where the preferentially formed oxides are in the liquid state [27] and the components of the passive layer and the electrolyte are incorporated into the PEO layer. This enables the modification of the layer composition, e.g., for the production of Al- or Si-rich oxide layers during the PEO of steel substrates in aluminate- or silicate-containing solutions with the aim of increased corrosion and/or wear resistance (summarized in [28]). The results of Li et al. showed that the initially formed surface layer, which contains Al and P, is converted into a porous PEO layer with a similar chemical composition by the first wave of microarc discharges [25]. With increasing duration, the layer thickness and the Al and Fe contents of the layer increase. This results in a PEO layer consisting of $\text{FeAl}_2\text{O}_4$ and $\text{Fe}_3\text{O}_4$ phases [25]. Other publications have described the production of PEO layers which largely consist of amorphous and crystalline $\text{Al}_2\text{O}_3$ phases using aluminate-containing electrolytes [19,21,29]. These PEO layers possess a very high hardness of up to 1680 HV and improve both the tribological behavior and the corrosion resistance of the steel substrate [29]. From the state-of-the-art research, it can be deduced that the formation of a surface layer of insoluble compounds such as aluminum oxide, hydroxide, and/or phosphate not only ensures the necessary substrate passivation, but also ensures that the top layer provides a significant portion of the chemical elements to be incorporated into the layer (e.g., Al), especially in the early phase of PEO. The formation of the top layer is therefore of central importance for the PEO of steels in aluminate electrolytes.

It can be shown from the Nernst equation that the anode potential for oxygen evolution decreases with increasing pH value and is well below 1 V in alkaline media. If the surface layer is formed by precipitation due to acidification at the anode, it can be expected that layer formation will already start in this potential range. The classic passivation by the formation of iron hydroxides or oxides takes place at anodic potentials of a few

100 millivolts. To our knowledge, polarization experiments in aluminate electrolytes in a potential range of up to 4 V have not yet been described in the scientific literature. Therefore, it is the aim of this paper to clarify whether, and if so, in which potential range, passivation- and pH-induced precipitation take place and which microstructural features characterize the layers. This approach allows to obtain novel findings, which enable a more precise control of the insulating layer formation prior to the PEO process.

## 2. Materials and Methods

### 2.1. Materials

The DP steel CR440Y780T-DP/HCT780XD (mass fraction in %: <0.17 C, <0.3 Si, <2.0 Mn, <0.05 P, <0.01 S, 0.015–0.08 Al, <1.0 Cr + Mo, <0.05 Nb + Ti), provided by Salzgitter Flachstahl GmbH, Germany, as hot-dip galvanized sheets with a thickness of about 1.7 mm, served as the substrate material. The samples were cut to a size of $15 \times 15$ mm$^2$ by water jet cutting. Subsequently, about 0.1 mm to 0.2 mm of the sheet thickness was removed by grinding on one side. In this way, the hot dip galvanizing coating (thickness < 10 µm) and an edge region with a slightly different metallographic appearance (thickness approx. 50 µm, possibly decarburized) were removed. Furthermore, a blank metal surface with a defined roughness of $R_a \approx 0.4$ µm and $R_z \approx 3.0$ µm (transverse to the grinding direction) was obtained by grinding. Directly before the electrochemical measurements, the samples were degreased with ethanol. The samples appeared metallically bright. Pickling was avoided in order not to preferentially dissolve electrochemically fewer noble phases and thus change the phase composition on the surface.

### 2.2. Electrochemical Polarization

The schematic set-up of the electrochemical polarization measurements is shown in the left of Figure 1. The DP steel sample was clamped in a cylindrical sample holder in a way that it is contacted on the back and masked on the front. The measurement area was about 78.5 mm$^2$ (circular opening with a diameter of 10 mm). A platinum foil with an area of about $15 \times 15$ mm$^2$ was used as the counter-electrode. Ag/AgCl/3M KCl served as the reference electrode. For the qualitative measurement of the pH value change during polarization, polarization experiments were carried out in a vertical electrode arrangement (schematically shown in Figure 1, right). The surface of the working electrode was also 78.5 mm$^2$, a platinum foil with an area of about $15 \times 15$ mm$^2$ served as the counter electrode, and Ag/AgCl/3M KCl was used as the reference electrode. In addition, a pH electrode EGA 133 (Sensortechnik Meinsberg, Waldheim, Germany) was positioned at a small angle at a distance of about 10 mm from the working electrode. The pH electrode was grounded via a high-impedance resistor.

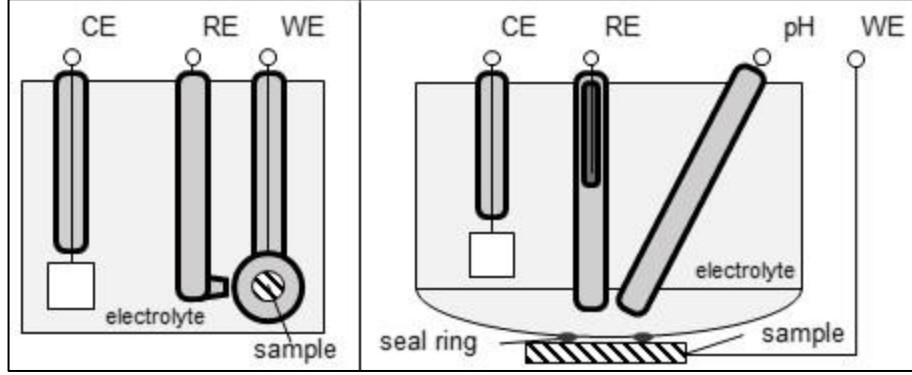

**Figure 1.** Schematic set-ups of the electrochemical measurements without (**left**) and with a pH electrode (**right**).

Table 1 gives an overview of the electrolytes used. The starting point was the results of Simchen et al., who observed the strongest substrate passivation in an aluminate electrolyte

at pH 12 and low phosphate content [19]. The aluminate content was set at 0.2 mol/L, as this causes a pH value of about 12 (electrolyte 1). The phosphate concentration was increased to 0.05 and 0.1 mol/L to investigate the influence of increased phosphate levels and the associated pH change. The electrolyte was synthesized by first quickly adding $NaAlO_2$ to stirred distilled water, as the aluminate addition itself causes the alkaline pH that is required for obtaining a stable solution. Afterwards, $Na_2HPO_4$ was added in the required amounts. In order to distinguish the effects of pH and phosphate concentration, reference electrolytes with pH 12 were also prepared by adding KOH. All chemicals were of analytical grade.

**Table 1.** Chemical composition and pH of the electrolyte solutions used in this study.

| Electrolyte No. | Concentration in mol/L | | pH |
| --- | --- | --- | --- |
| | $NaAlO_2$ | $Na_2HPO_4$ | |
| 1 | 0.2 | 0 | 12.0 |
| 2 | 0.2 | 0.05 | 11.7 |
| 3 | 0.2 | 0.05 | 12.0 * |
| 4 | 0.2 | 0.1 | 11.5 |
| 5 | 0.2 | 0.1 | 12.0 * |

* pH adjusted to 12 by adding KOH.

Before each electrochemical measurement, the open-circuit potential (OCP) was first recorded for 30 min. For all electrolytes, potentiodynamic "screening" measurements (OCP − 100 mV to OCP + 4000 mV with a scan rate of 10 mV/s), as well as higher resolution measurements near the OCP (OCP − 100 mV to OCP + 100 mV with 1 mV/s) were performed. Based on these higher resolution measurements, the corrosion potential, $\varphi_{corr}$, was determined and the corrosion current density, $j_{corr}$, was calculated according to the method of Stern [30] using Equations (12) and (13), where $R_{pol}$ is the polarization resistance, $j$ is the current density, $R$ is the universal gas constant, $T$ is the electrolyte temperature (295.15 K), $F$ is the Faraday constant, and $A$ is the measurement area.

$$R_{pol} = \frac{(\varphi_{corr} + 10 \text{ mV}) - (\varphi_{corr} - 10 \text{ mV})}{I(\varphi_{corr} + 10 \text{ mV}) - I(\varphi_{corr} - 10 \text{ mV})} \tag{12}$$

$$j_{corr} = \frac{R \cdot T}{z \cdot F \cdot R_{pol} \cdot A} \tag{13}$$

Additionally, potentiostatic polarization measurements were carried out for all electrolytes at OCP + 4 V to produce thick surface layers for further material and scientific characterization, as well as further potentiostatic measurements at defined potentials. For statistical validation, all electrochemical measurements were carried out at least 3 times under the same conditions. The electrochemical work station Zennium X (Zahner, Kronach, Germany) served as the voltage source and for recording the current curves and the pH value.

*2.3. Microstructural Analysis*

All specimens were routinely documented using a stereo microscope MVX 10 (Olympus, Tokyo, Japan). Metallographic preparation was carried out on samples that were polarized at an anodic potential of OCP + 4 V. For this purpose, the samples were cut, embedded in conductive resin, ground on SiC paper to 4000 grit, polished on cloths to a diamond size of 1 μm, and finally polished with a suspension of colloidal silicon dioxide. The optical microscopic examinations were carried out using an inverse optical microscope GX 51 (Olympus, Japan) in bright field mode. Prior to scanning electron microscopy (SEM) measurements, the cross-sections were rinsed in ethanol and isopropanol and then dried in an oven at 60 °C. In order to ensure a sufficient electrical conductivity of the electrically insulating layers for the SEM investigations, all ground surfaces were vapor coated with carbon. The scanning electron microscopic investigations were carried out with an SEM

LEO1455VP (Zeiss, Oberkochen, Germany) at an acceleration voltage of 25 kV and a working distance of 14.5 mm using the secondary electron (SE) and backscattered electron (BSE) contrasts. In addition, the chemical composition of the surface layers was determined using energy-dispersive X-ray microanalysis (EDX) in the middle of the layer.

In order to determine the phase composition, an X-ray diffraction (XRD) analysis was carried out using the D8 Discover (Bruker, USA) diffractometer with Co-Kα radiation. The measurements were performed on the surface of samples that were polarized at OCP + 4 V with a point focus (diameter pinhole aperture 0.5 mm) and the LYNXEYE XE-T detector. Measurements for the qualitative determination of the phase composition were carried out on the same sample surfaces using a confocal Raman microscope inVia (Renishaw, United Kingdom). The measurement was carried out with a 20× lens and a laser wavelength of 532 nm at 100% excitation energy for 10 s (thick precipitated layers) or 50 s (thin passive layers) with 10 accumulations. The reference data of possible phases were taken from the RRUFF database [31] and from the scientific literature.

## 3. Results

### *3.1. Polarization Experiments*

#### 3.1.1. Potentiodynamic Polarization

In order to obtain an overview of the electrochemical processes, potentiodynamic polarization measurements were first carried out in the potential range from OCP − 100 mV to OCP + 4000 V. Figure 2 shows the average current–density curves (solid lines) including the curves of the standard deviations of the current–density values at a specific potential for different electrolytes in the range between $\varphi_{corr}$ and OCP + 4 V, as well as a representative curve of the pH change during polarization in electrolyte 1. The secondary ordinate (pH change) has a linear scale and is not labeled with absolute values of the pH change, since the pH measurements were not carried out in the immediate vicinity of the anode but at a distance of about 10 mm. It is to be expected that the pH change in the immediate vicinity of the substrate will be significantly higher than measured.

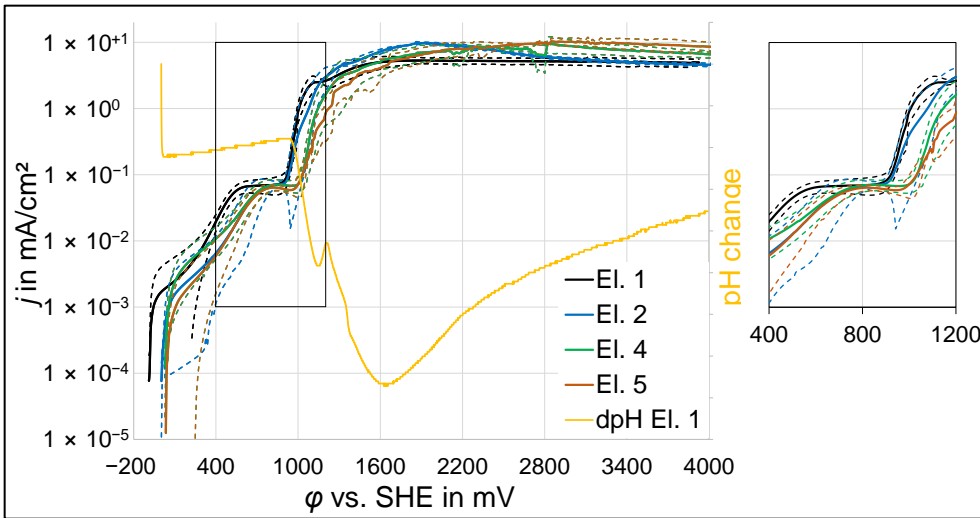

**Figure 2.** $j$–$\varphi$ curves due to potentiodynamic polarization in electrolytes (El.) 1, 2, 4, and 5 within the range of $\varphi_{corr}$ to OCP + 4 V with a magnified image of the section between 400 and 1200 mV vs. SHE (right) and qualitative pH change (dpH) during a measurement in electrolyte 1 (linear scale). The dashed lines represent the standard deviation between the repetition samples.

The current–density curves are qualitatively similar for all electrolytes. After a steep increase when crossing $\varphi_{corr}$, the curves flatten slightly at about $\varphi_{corr}$ + 10 mV. In the region of the steep current–density increase, a sudden pH decrease can be observed. Another steep current–density increase can be identified starting at about $\varphi_{corr}$ + 350 mV. Again, a

flattening of the curves can be seen at around $\varphi_{corr}$ + 600 mV (electrolyte 1) and at around $\varphi_{corr}$ + 700 mV (other electrolytes). In the case of electrolytes 4 and 5 (highest phosphate concentration of 0.1 mol/L), there is even a slight decrease in current density and a local minimum is passed (see magnified section in Figure 2). Another steep increase in current density follows at around 900 mV vs. standard hydrogen electrode (SHE) for electrolytes 1 and 2 and at around 1000 mV vs. SHE for the other electrolytes. Compared to the current–density increase, the pH starts to decrease during polarization in electrolyte 1 with a slight delay at around 950 mV vs. SHE. The pH value reacts very sensitively to a change in current density. For example, the current–density fluctuation during polarization in electrolyte 1 at around 1200 mV vs. SHE leads to a measurable pH fluctuation. The increase in current density is the steepest for electrolyte 1 and begins to level off at around 1000 mV vs. SHE. After passing through the minimum pH value at around 1650 mV vs. SHE, the maximum current density of 5.3 mA/cm$^2$ is reached for electrolyte 1 at around 2000 mV vs. SHE. The flattening of the curves and the current–density maxima shift towards higher potentials with increasing phosphate content in the electrolyte. In addition, the magnitudes of the maxima increase to around 9.5–10.5 mA/cm$^2$. During polarization in electrolyte 5 (high phosphate content, adjusted to pH 12), the highest measured current–density maximum is reached at about 2850 mV vs. SHE. At the end of the polarization at OCP + 4 V, the lowest current densities in the range of 4.5 mA/cm$^2$ to 5 mA/cm$^2$ are reached for electrolytes 1 and 2. After polarization in the potential range OCP − 100 mV to OCP + 4000 mV, light gray to white covering layers can be observed on all sample surfaces with the naked eye.

As can be seen in Figure 2, different free corrosion potentials are obtained in the different electrolytes. The OCP drifts significantly during the 30 min OCP measurement prior to the potentiodynamic polarization measurements. Both increases and decreases in the OCP were measured in the same electrolyte. A detailed investigation of the electrochemical behavior was carried out using polarization tests in the OCP ± 100 mV range. Figure 3 shows the courses of the current density for the polarization in electrolytes 1, 3, and 5 (which have the same pH value of 12 and differ in terms of the phosphate content) in the potential range $\varphi_{corr}$ ± 40 mV using a Tafel plot. The points marked with the symbol × represent the corrosion potential, $\varphi_{corr}$, and the corrosion current density, $j_{corr}$, calculated according to Equation (13). The average values, including standard deviations of $\varphi_{corr}$ and $j_{corr}$, are summarized for all electrolytes in Table 2.

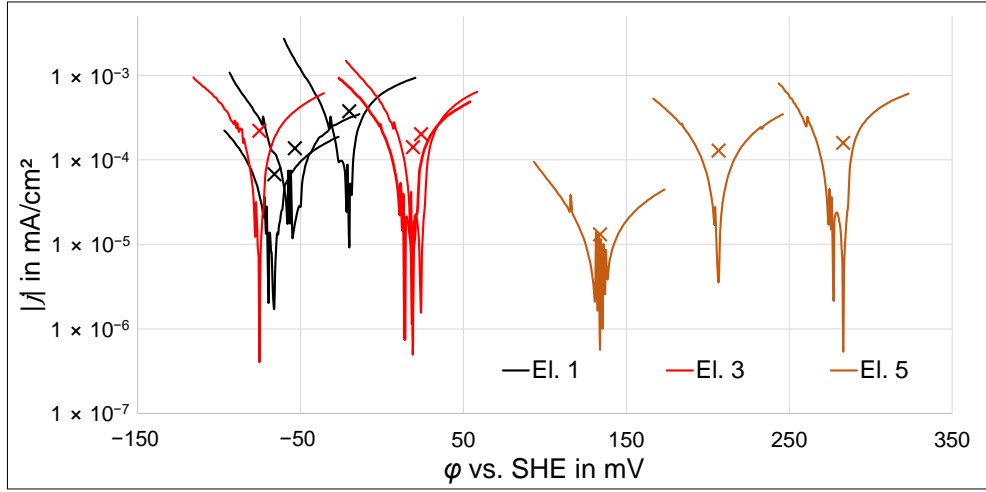

**Figure 3.** $j$–$\varphi$ curves in electrolytes no. 1, 3, and 5 within the range of $\varphi_{corr}$ ± 40 mV; the crosses mark the $\varphi_{corr}$ and $j_{corr}$ values of every curve.

The most negative corrosion potentials were measured for the electrolytes without phosphate and with 0.05 mol/L phosphate. No clear influence of the pH value on the corrosion potential can be identified for these electrolytes when considering the large standard deviations. In contrast, for electrolytes 4 and 5 with a phosphate concentration of

0.1 mol/L, a shift in the corrosion potential towards more positive values was observed (see Table 2), especially after adjustment of the pH value to 12 (electrolyte 5). The corrosion current density tends to be lower for electrolytes 4 and 5 compared to electrolytes 1, 2, and 3. After the polarization tests in the OCP $\pm$ 100 mV range, no macroscopic changes or layers can be seen on the sample surface.

**Table 2.** Average values and standard deviations of $\varphi_{corr}$ and $j_{corr}$ (Equation (13)) in different electrolytes.

| Electrolyte No. | $\varphi_{corr}$ in mV | $j_{corr}$ in $10^{-5} \cdot$ mA/cm$^2$ |
|---|---|---|
| 1 | $-47 \pm 24$ | $19 \pm 16$ |
| 2 | $-70 \pm 150$ | $13 \pm 7$ |
| 3 | $-10 \pm 60$ | $19 \pm 5$ |
| 4 | $90 \pm 80$ | $9 \pm 7$ |
| 5 | $210 \pm 80$ | $10 \pm 8$ |

### 3.1.2. Polarization at Constant Potential

The layer formation was examined in more detail using potentiostatic polarization experiments at OCP + 4 V. As can be seen from Figure 4, the current–density curves for electrolytes 1, 2, and 3 are very similar and differ only within the range of their standard deviations. They can be described to a very good approximation ($R^2 \approx 0.99$) by power functions of the form $j = a \cdot t^b$, with $a$ between 33 mA/(cm$^2 \cdot$s) and 45 mA/(cm$^2 \cdot$s) and $b$ between $-0.43$ and $-0.47$.

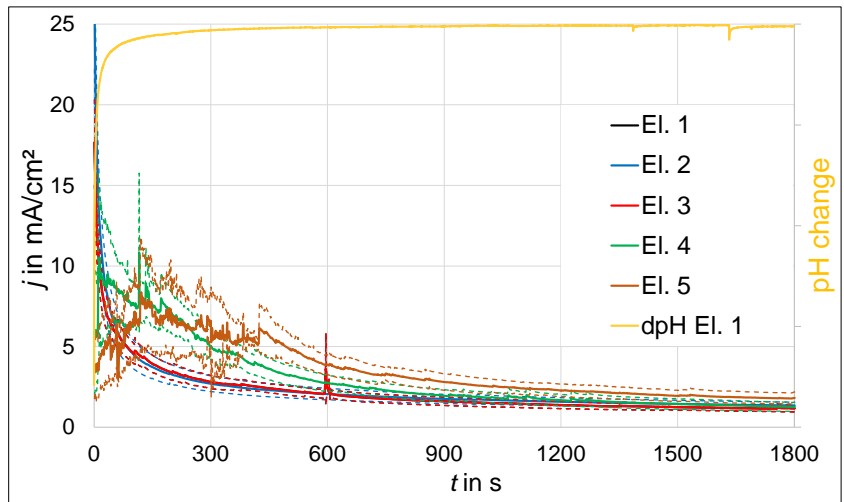

**Figure 4.** $j$–$t$ curves due to polarization at constant potential (OCP + 4 V) in the different electrolytes and qualitative pH change (dpH) during a measurement in electrolyte 1. The dashed lines represent the standard deviation between the repetition samples.

In addition to a high current density at the beginning of the polarization, a considerable decrease in the pH value from 12 to 8.5 can be observed at a distance of about 10 mm from the anode, which is a pH change of $-3.5$ within the first seconds. With the rapid decrease in current density, the pH value quickly approaches the original pH value of the electrolyte. When polarizing in electrolyte 4 and in particular in electrolyte 5, higher current densities and stronger current–density fluctuations can be seen shortly after the start. Later during the experiment, a continuous decrease in the current density can also be observed for these electrolytes. After 1800 s polarization in electrolytes 1 to 4, the average current densities are about 1.17 mA/cm$^2$ to 1.35 mA/cm$^2$. These differences are still within the standard deviation of the individual curves. At the same time, a higher average current density of about 1.8 mA/cm$^2$ can be measured at the end of the polarization in electrolyte 5.

The surface layers after 30 min of polarization at OCP + 4 V have a light gray to white color and are partially opaque so that the grinded steel substrate is visible. In Figure 5, stereomicroscopic surface images of one representative sample per electrolyte are arranged in a table structure according to the phosphate content of the electrolyte (e.g., "0.05 $PO_4^{3-}$" stands for 0.05 mol/L phosphate) and pH value. The images were always taken at the center of the measuring surface. After polarization in electrolyte 1 (0 $PO_4^{3-}$, pH 12), a microscopically heterogeneous top layer is formed. There are finely distributed light gray, opaque areas next to fine pores (darker areas) where the steel substrate shines through the layer more strongly. The top layers produced in electrolytes 2 (0.05 $PO_4^{3-}$, pH 11.7) and 4 (0.1 $PO_4^{3-}$, pH 11.5) appear homogeneous and less porous. In the case of the latter layer (0.1 $PO_4^{3-}$, pH 11.5), sharp-edged layer spallation can be seen, which indicates a high degree of brittleness. After polarization in electrolytes 3 (0.05 $PO_4^{3-}$, pH 12) and 5 (0.1 $PO_4^{3-}$, pH 12), larger pores are visible on the surface, which are presumably caused by the temporary adhesion of gas bubbles and the associated hindrance of layer formation. This is most pronounced after polarization in electrolyte 5.

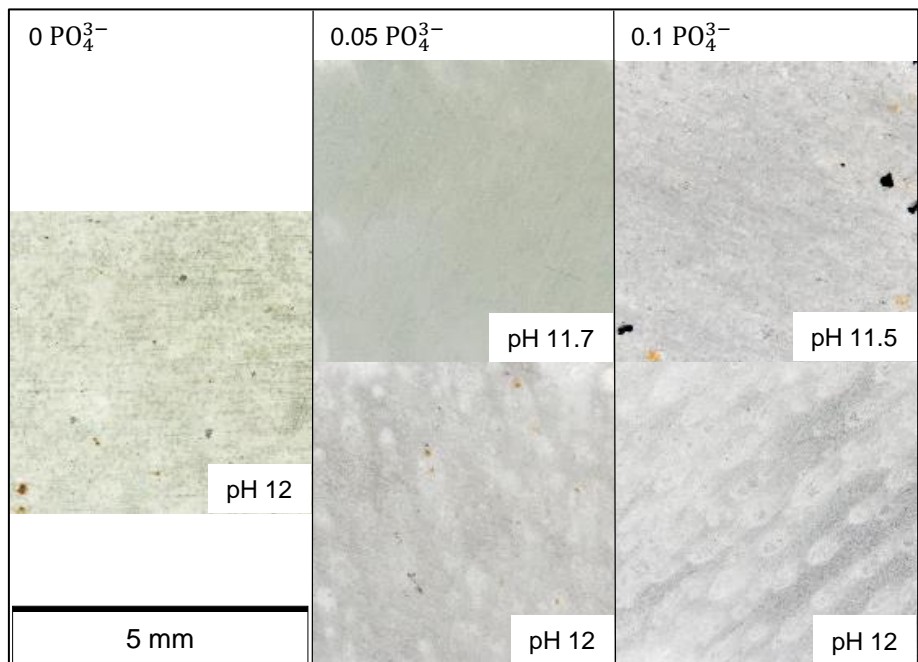

**Figure 5.** Optical microscopic images of the sample areas after polarization at constant potential (OCP + 4 V) in the different electrolytes, categorized by the phosphate content in the electrolyte ($PO_4^{3-}$) in mol/L and the pH. The scale bar applies to all images.

When examining the surfaces at higher magnification using optical microscopy, it is noticeable that all layers are microcracked. After polarization in electrolytes 1, 2, and 3, a fine network of closed cracks can be observed. As shown in the left of Figure 6, the layer, which was produced by polarization in electrolyte 2 and appeared very evenly in Figure 5, also has fine porosity and roughness. This may cause light scattering, which is the reason why the layer appears cloudy and the substrate cannot be seen. For the same reason, the crack network can only be seen in some places (see Figure 6, left). Dense crack networks are clearly visible in layers that were created by polarization in electrolytes 4 and 5. This is particularly pronounced after polarization in electrolyte 4. As can be seen from the right of Figure 6, some cracks have widened so much that layer fragments are present as islands. In addition, these layers are optically transparent so that the grinding marks on the base material can be clearly seen in the background. The macroscopic gray appearance may not be due to fine porosity or roughness but because of light scattering at the crack edges.

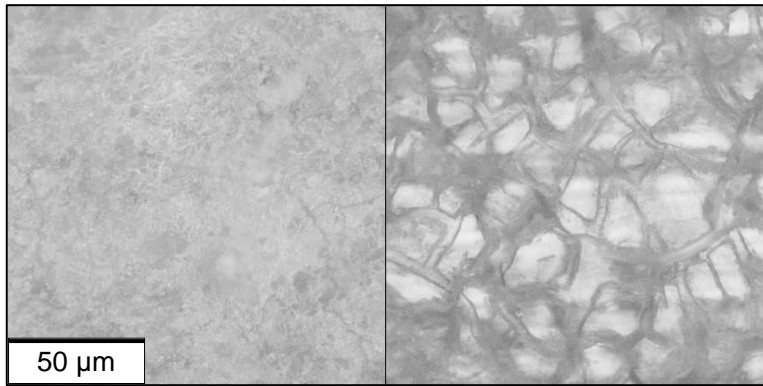

**Figure 6.** Optical microscopic images of the sample surface after polarization at constant potential (OCP + 4 V) in electrolytes 2 (**left**) and 4 (**right**). The scale bar applies to both images.

As already described in Section 3.1.1, the potentiodynamic polarization experiments depicted in Figure 2 show a flattening of the current–density curve in the potential range between about 550 mV and 1000 mV vs. SHE (depending on the electrolyte), and sometimes there is even a slight decrease in current density. To examine this potential range more closely, potentiostatic polarization tests were carried out at OCP + 500 mV. Figure 7 shows the current–density curves for electrolytes 1 and 4, which differ most in terms of phosphate content and pH value. In both cases, a clear decrease in current density is recorded shortly after the start. Compared to the polarization at OCP + 4 V (Figure 4), the current densities after 1800 s are about 20 to 30 times higher. Up to about 1100 s, both average current–density curves (solid lines) run almost congruently.

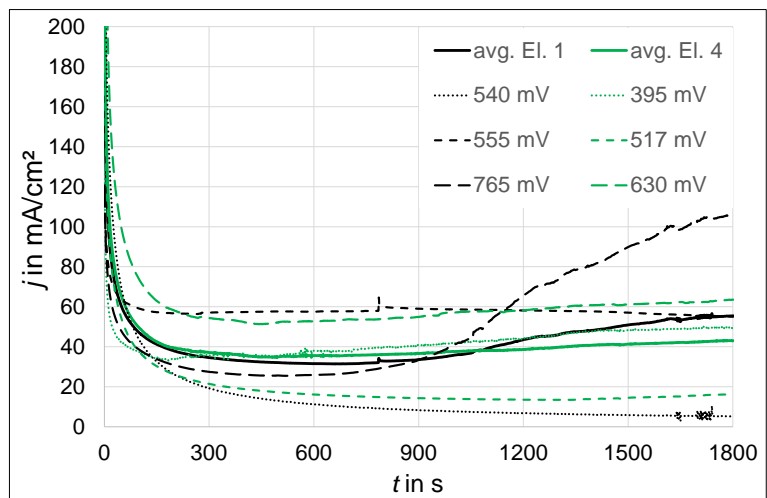

**Figure 7.** $j$–$t$ curves due to polarization at constant OCP + 500 mV in electrolytes 1 and 4. The solid lines represent the averaged curves; the dashed lines represent the curves of the individual samples with the potentials of the individual measurements vs. SHE.

However, it can be observed that the courses of the individual curves in Figure 7 are very different. It must be taken into account that the individual measurements were carried out at different potentials vs. SHE, since the OCPs varied considerably. The potentials vs. SHE of the individual measurements are given in the legend of Figure 7. For both electrolytes, the lowest current densities of about 4 mA/cm$^2$ for electrolyte 1 and about 13 mA/cm$^2$ for electrolyte 4 were reached after 1800 s polarization at 540 mV (electrolyte 1) and 517 mV (electrolyte 4) vs. SHE. These values are close to the oxygen evolution potentials of 522 mV and 552 mV vs. SHE at pH 12 and 11.5, respectively. Relatively high current densities of over 50 mA/cm$^2$ were observed both above 550 mV vs. SHE and below 400 mV vs. SHE. At the highest potential of 765 mV vs. SHE, a significant increase in the

current density was observed at the end of the measurement. After the tests, no surface layer formation can be seen macroscopically or by means of optical microscopy.

*3.2. Microstructure*

To illustrate the layer microstructure, Figure 8 shows optical microscopy images of the layer cross-section of a representative sample of each electrolyte. The arrangement of the images is the same as in Figure 5. In accordance with the observation of pores in the surface images, significant layer thickness fluctuations can be observed in the layer cross-section. However, it cannot be clearly assigned which regions with a thin layer actually formed during polarization. The surface layers showed a high degree of brittleness and low adhesion, which led to partial layer detachment during the cross-section preparation. The amount of cracks visible in Figure 8 and the gaps between the layer and the substrate probably increased during preparation. Layer thicknesses of about 10 μm and 20 μm within the uniform regions (shown in Figure 8) were probably only slightly changed by the preparation of the cross-section. These uniform layer areas have a similar appearance in all cases.

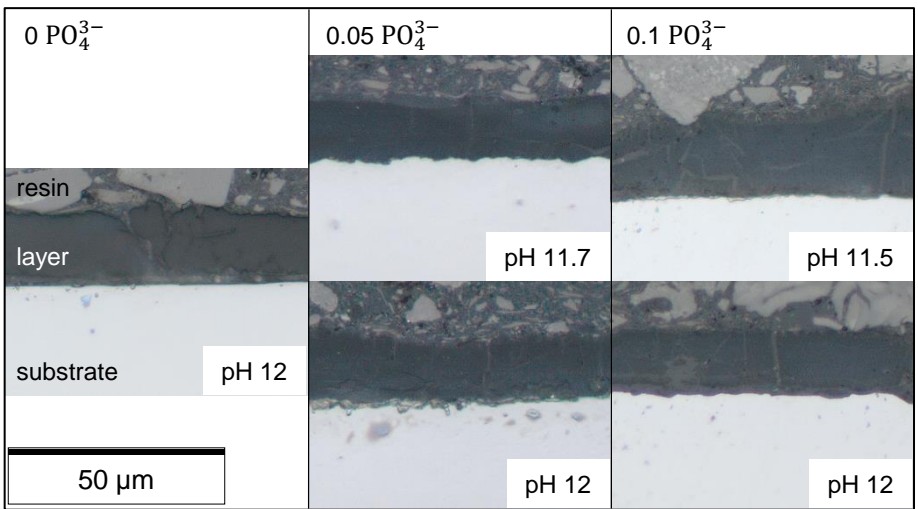

**Figure 8.** Optical microscopic images of the cross-sections after polarization at constant potential (OCP + 4 V) in the different electrolytes, categorized by the phosphate content in the electrolyte ($PO_4^{3-}$) in mol/L and the pH. The scale bar applies to all images.

Generally, the same layer characteristics can be recognized by means of SEM. Figure 9 shows homogeneous layer regions that were created by polarization at OCP + 4 V in electrolytes 1, 3, and 5 with different phosphate concentrations and at the same pH of 12. These layer areas still appear compact in the SE contrast. In the BSE contrast, a very faint horizontal, slightly wavy layering can be seen at some places. This is an indication of slight variations in the chemical composition. However, a systematic change in the composition over the layer thickness is not recognizable. In order to determine representative average values of the chemical element content, EDX measurements were always carried out over a larger measuring area in the middle of the layer, with a sufficient distance to the edges. The results of the EDX measurements are summarized in Table 3. Polarization in electrolyte 1 (without phosphate) creates layers that almost exclusively consist of the elements Al and O, with the ratio of Al to O being about 2 to 3. With the addition of 0.05 mol/L and 0.1 mol/L phosphate to the electrolyte, the P content of the layers increases to an average molar fraction of 7.9% and 9.3%, respectively. At the same time, the O content increases from 61.4% to 66.4% and 68.7%, respectively. These increases are at the expense of the Al content, which drops significantly from 38.4% to 25.0% and 20.5%, respectively. In any case, the element Fe can only be measured in small amounts. It cannot be clearly determined whether the layer actually contains Fe. It is also conceivable that Fe particles

were transferred to the layer during the preparation of the cross-section or that the substrate was slightly excited during the EDX measurement.

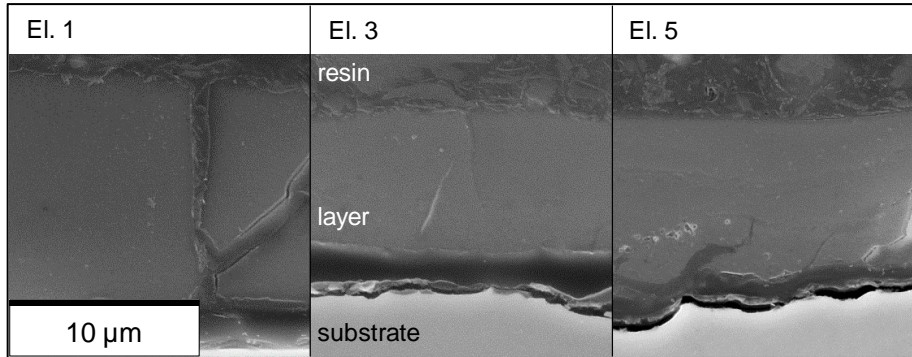

**Figure 9.** SEM images (SE contrast) of the cross-sections of samples that were polarized at constant OCP + 4 V in electrolytes 1, 3, and 5. The scale bar applies to all images.

**Table 3.** Chemical compositions measured by EDX analyses at the cross-sections of samples that were polarized at constant potential (OCP + 4 V) in electrolytes 1, 3, and 5.

| Electrolyte No. | Molar Fraction in % | | | |
|---|---|---|---|---|
| | **Al** | **O** | **P** | **Fe** |
| 1 | $38.4 \pm 0.8$ | $61.4 \pm 0.7$ | <0.1 | $0.3 \pm 0.1$ |
| 3 | $25.0 \pm 1.9$ | $66 \pm 3$ | $7.9 \pm 0.9$ | $0.8 \pm 0.2$ |
| 5 | $20.5 \pm 0.7$ | $68.7 \pm 2.1$ | $9.3 \pm 0.1$ | $1.5 \pm 1.3$ |

Despite the high layer thicknesses in the range of 10 µm to 20 µm after polarization at OCP + 4 V for 30 min, only the characteristic peaks of the substrate material can be registered with XRD surface measurements. Obviously, the layers are X-ray amorphous or nanocrystalline.

The surface layers that were generated by polarization at OCP + 4 V for 30 min in electrolytes 1 and 4, which differ most in terms of phosphate content and pH value, were analyzed using Raman spectroscopy. As can be seen from the bottom spectrum in Figure 10, the layer produced in electrolyte 1 primarily shows a broad peak with a maximum at 590 cm$^{-1}$ and a double peak at around 1070 cm$^{-1}$ and 1100 cm$^{-1}$. Based on the results of the EDX measurements, only phases containing Al, O, and possibly light elements such as H, which are not detectable by EDX, can be considered. Suitable reference spectra of the Al(OH)$_3$ modifications gibbsite (RRUFF ID: R190038) and nordstrandite (RRUFF ID: R050592) were taken from [31] and inserted in Figure 10. The pronounced gibbsite band at about 475 cm$^{-1}$ and the weak band at about 1080 cm$^{-1}$ may be included in the broad peaks of the electrolyte 1 spectrum, but do not explain them sufficiently. The nordstrandite reference spectrum contains a series of bands between 470 cm$^{-1}$ and 750 cm$^{-1}$. In the case of an amorphous or nanocrystalline layer, these bands could appear broader and overlap to form a broad peak as measured at the sample produced in electrolyte 1. However, the other bands of the nordstrandite spectrum between 210 cm$^{-1}$ and 450 cm$^{-1}$ and between 810 cm$^{-1}$ and 1000 cm$^{-1}$ do not correspond to the measured spectrum. It is known from the literature that the Al(OH)$_3$ modification bayerite has characteristic bands at the following Raman shifts (in cm$^{-1}$): 1079, 1068, 898, 866, 545, 525, 484, 443, 434, 388, 359, 322, 297, 250, 239, and 205 [32]. These also do not specifically match the bands of the measured spectrum of the layer produced in electrolyte 1. Furthermore, there is no striking conformance with boehmite (RRUFF ID: R120123) either, which is characterized by bands at around 360 cm$^{-1}$, 500 cm$^{-1}$, and 680 cm$^{-1}$ [31]. According to Sudare et al., the Raman spectra of amorphous Al$_2$O$_3$ and γ-Al$_2$O$_3$ show bands at about 555 cm$^{-1}$ and 1060 cm$^{-1}$ within the applied measurement range (vertical, black lines in Figure 10) [33]. Both bands are close to the characteristic bands of the spectrum of the layer produced in electrolyte 1.

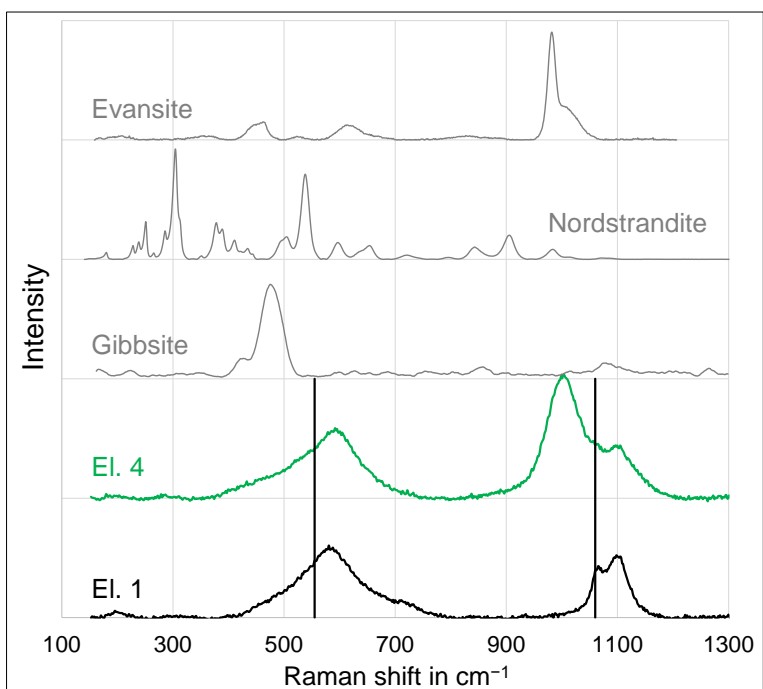

**Figure 10.** Raman spectra of layers produced in electrolytes 1 and 4 at OCP + 4 V and reference spectra of gibbsite, nordstrandite, and evansite from [31]. The vertical black lines mark characteristic bands of amorphous $Al_2O_3/\gamma\text{-}Al_2O_3$ according to [33].

Compared to the layer from electrolyte 1, the layer produced in electrolyte 4 primarily shows an additional broad peak at about 1000 cm$^{-1}$. In addition, the left shoulder of the broad peak with the maximum at 590 cm$^{-1}$ extends more towards lower values. Based on the results of the EDX measurements, the conformability of phases containing P in addition to Al, O, and potentially H was checked. The best match was found for the evansite phase. This is a hydrous phosphate that also contains Al ions and is described by the chemical formula $Al_3(PO_4)(OH)_6\cdot6H_2O$. As can be seen in Figure 10, the deviations between the spectra of the layers of electrolytes 1 and 4 can be explained very well by the characteristic bands in the Raman spectrum of Evansite at around 980 cm$^{-1}$ and 460 cm$^{-1}$. The broad peak around 590 cm$^{-1}$ possibly covers another band of Evansite at around 620 cm$^{-1}$. In order to clarify the phase composition of surface layers, which are generated during polarization at OCP + 500 mV, the individual samples from electrolytes 1 and 4, which showed the strongest current–density decrease during polarization, were examined using Raman spectroscopy as well. A five times higher measurement duration was applied in order to obtain peaks which can be clearly distinguished from the background. As can be seen in Figure 11, the Raman spectra of the samples produced in electrolytes 1 and 4 are very similar. They essentially show the same peaks, but the latter spectrum shows higher peak intensities at 1130 cm$^{-1}$, 1290 cm$^{-1}$, and 1440 cm$^{-1}$. Both spectra differ significantly from the spectra of the macroscopically visible layers that were produced by polarization at OCP + 4 V. A Raman spectrum with the same peaks but lower intensities was measured at the edge of a steel sample. This area was not polarized in either electrolyte but subjected to the same rinsing routine and storage. For phase identification, the reference spectra of phases containing Fe, O, and potentially H were checked. By far the best match was found for the maghemite phase ($\gamma\text{-}Fe_2O_3$) in [34]. As can be seen in Figure 11, there is a high level of agreement with the reference spectrum (RRUFF ID: R140712) from [31] for almost all peaks in the range between about 200 cm$^{-1}$ and 1300 cm$^{-1}$, with the exceptions that the measured spectra exhibit a much more pronounced peak at about 770 cm$^{-1}$ and no peaks around 500 cm$^{-1}$. The measured peaks above 1300 cm$^{-1}$ were compared with references of the maghemite phase from the literature. Hanesch et al. identified a characteristic band at 1330 cm$^{-1}$ [35]. This agrees approximately with the findings of Mazzetti and

Thistlethwaite, according to whom there is a band at 1320 cm$^{-1}$ [36]. In addition, Mazzetti and Thistlethwaite identified a band at 1560 cm$^{-1}$ [36]. These bands are shown as black lines in Figure 11 and are close to the measured peaks at 1290 cm$^{-1}$ and 1550 cm$^{-1}$. Furthermore, Modesto Lopez et al. reported a broad band between about 1360 cm$^{-1}$ and 1480 cm$^{-1}$ [37]. The measured peak at 1435 cm$^{-1}$ is located within this range.

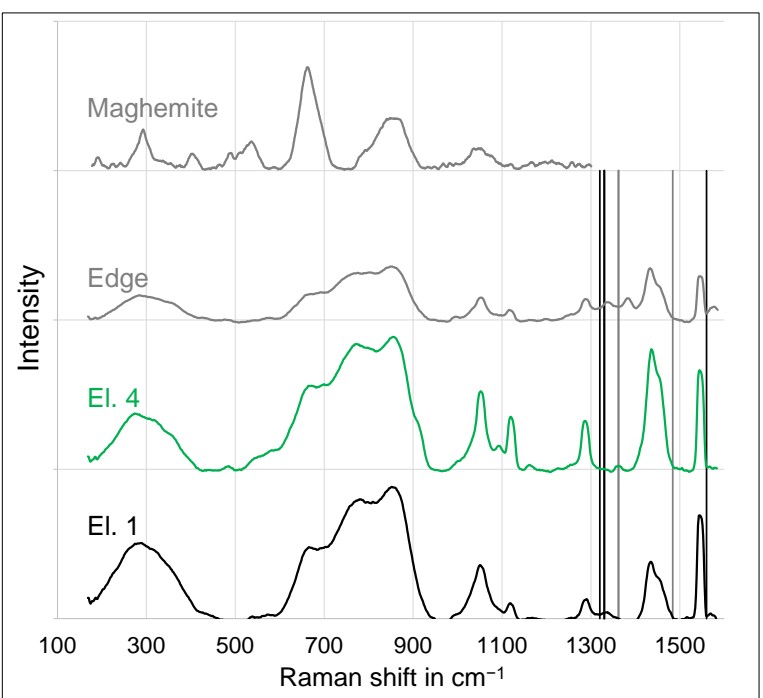

**Figure 11.** Raman spectra of layers produced in electrolytes 1 and 4 at OCP + 500 mV and reference spectra of the untreated sample edge and maghemite from [31], the vertical lines mark characteristic bands (black) and a broad peak (gray) of maghemite according to the literature [35–37].

## 4. Discussion

The results of the polarization experiments indicate that several different processes take place on the substrate surface in the potential range from $\varphi_{corr}$ to OCP + 4 V. This approach provides novel insights into the electrochemical behavior of steel in alkaline aluminate solution. The fluctuations during the OCP measurements and the different OCPs of individual samples in the same electrolyte appear to be stochastic and must therefore be caused by randomly varying factors. One reason could be the multi-phase microstructure of the DP steel, which essentially consists of ferrite and martensite. The microstructure shows a slight banding parallel to the rolling direction. When grinding the surface, a line rich in the electrochemically more noble martensite or the less noble ferrite may randomly be exposed, resulting in different OCP values and a more or less pronounced drift of the OCP due to the passivation of the less noble ferrite phase. The corrosion potential of $-47 \pm 24$ mV vs. SHE, measured in electrolyte 1, is in good agreement with the corrosion potential of around $-56$ mV vs. SHE of a DP steel in 0.8 mol/L NaOH reported in the literature [17]. A corrosion-inhibiting effect in alkaline media (pH 12 to 13) due to the adhesion of phosphates to the steel surface is described in the literature [38]. This could explain the slightly lower average corrosion current densities at the highest phosphate concentration. The interactions between the phosphate ions and the steel surface are not described in the literature in detail.

Since the potentiodynamic polarization experiments always started in the cathodic region at OCP − 100 mV, in order to reliably polarize slightly cathodically when starting the measurement despite the OCP fluctuation over time, adhering anions were probably initially repelled. For this reason, the OCPs after 30 min of immersion do not exactly

match the $\varphi_{corr}$ values of the potentiodynamic polarization experiments. Immediately after passing $\varphi_{corr}$, the current density initially increases sharply, which is probably due to the anodic dissolution of the iron. The subsequent flattening of the current–density curve might be explained by the formation of an iron(II) oxide or hydroxide layer. According to the cyclic voltammetric investigations by Joiret et al., this layer formation is associated with the formation of hydronium ions according to Equations (1) and (2) [7]. This might explain the decrease in pH measured after passing $\varphi_{corr}$. This layer is locally destroyed with increasing anodic potential, so the current density increases sharply again.

The subsequent current–density plateau between about 600 mV and 900 mV vs. SHE for the polarization in electrolytes 1 and 2 and between about 780 mV and 1000 mV vs. SHE for the polarization in electrolytes 4 and 5 (see Figure 2) indicates formation of another layer. Joiret et al. state that iron(II) oxide and hydroxide are firstly oxidized to iron(II,III) oxide (magnetite) according to Equations (3) and (4) and then further oxidized to iron(III) oxide and hydroxide according to Equations (5) and (6) [7]. It is known from the literature that a protective passive layer is only observed after the formation of iron(II,III) oxide [8,9]. This coincides with the observation that the flattening of the increase in current density is much more pronounced here.

Using potentiostatic polarization experiments in the vicinity of this potential range, it was possible to demonstrate the formation of a passive layer. The greatest decrease in current density over time was found at a potential of 500 mV vs. SHE, independent of the phosphate content and pH value of the electrolyte. This is a few 100 mV before the current–density plateau was reached in the potentiodynamic measurements, which can be explained by the kinetics of oxide formation. As can be seen in Figure 5, the current density drops quickly at the beginning of the polarization. As an example, it still takes around 30 s before the current density falls below twice the value of the current density minimum. Given the potential scan rate of 10 mV/s in potentiodynamic polarization up to OCP + 4 V, the anodic potential increases by 300 mV during this time.

On the measuring surfaces of the samples, which showed the strongest current density decrease during polarization at OCP + 500 mV, no phase other than maghemite could be detected by means of Raman spectroscopy. This also means that both other iron oxides or hydroxides and chemical compounds containing P or Al are not present in detectable amounts. The formation of maghemite ($\gamma$-$Fe_2O_3$) by oxidation of the iron(II,III) oxide according to Equation (5) is plausible [7]. Both the differences between the spectra of the samples, which were polarized in electrolytes 1 and 4, and the differences between the measured spectra and the maghemite reference could not be attributed to plausible phases. Since a Raman spectrum with the same characteristic peaks was measured at the edge of the sample that was not in contact with the electrolyte, it cannot be definitively proven whether the maghemite layer formation was actually due to polarization or as a result of sample storage. However, it might be concluded from the higher peak intensities that slightly thicker oxide layers were formed in the polarized areas. The layers cannot be distinguished from the substrate neither macroscopically nor under the optical microscope. This is consistent with the literature, which states that the thickness of an oxide layer formed in this potential range on iron is only a few nanometers [11].

According to the Nernst equation, oxygen evolution starts at 522 mV and 552 mV vs. SHE at pH 12 and 11.5, respectively. It is possible that this impairs the formation of dense layers during potentiostatic polarization and leads to the detachment of porous and non-adherent top layers. This would explain that at potentials above about 550 mV vs. SHE, a less pronounced decrease in current density could be measured and that in case of the highest potential of 765 mV vs. SHE, the current even increased again towards the end. In the case of the potentiodynamic measurements, the oxygen evolution appears with a significant delay at around 900 mV vs. SHE or 1000 mV vs. SHE, which can be seen by the significant increase in the current density and the significant decrease in the pH value that is delayed by about another 50 mV. It is possible that the passive layer initially inhibited oxygen evolution until it finally detaches from the surface. As the comparisons of the current–density curves for electrolytes 1

and 2 and in particular for electrolytes 4 and 5 in Figure 2 shows, this breakthrough potential is not significantly influenced by the pH value of the solution. The increased breakdown potential in electrolytes 4 and 5 with the highest phosphate content is possibly due to an interaction of the attached phosphate with the surface.

In the case of potentiodynamic polarization in electrolyte 1, the increase in current density levels off immediately after the start of the pH value reduction. These observations are consistent with the theory that a precipitation reaction according to Equations (9) or (10) takes place as a result of the pH drop and that the precipitated layer inhibits the current. Similarly, layer formation in phosphate-containing electrolytes at potentials above 1 V vs. SHE can be explained by the precipitation reaction described in Equation (11). During potentiodynamic polarization in phosphate-containing electrolytes, the maximum current density is reached at higher potentials and is at a higher current density level. Possibly, the development of oxygen is initially slightly inhibited by phosphate adhering to the anode. However, the precipitation layer appears to be more permeable to the released oxygen, resulting in the formation of more and bigger oxygen bubbles, which adhere to the surface temporarily. This coincides with the observation that after polarization at OCP + 4 V in electrolyte 5 (highest phosphate content, pH 12), the layer surface shows the most pronounced porosity due to gas evolution.

It can be expected that for any anodic potential between about 1 V and several hundred V, just below the breakthrough potential of microarc ignition, layer formation occurs due to a precipitation reaction, which is consistent with the findings of Li et al. [25]. At an anodic potential of OCP + 4 V, a pH reduction of up to 3.5 could be measured in electrolyte 1 at a distance of about 10 mm from the anode. A significantly greater reduction in the pH value is to be expected directly at the anode surface. This could at least partially invalidate the argument formulated in [20] that there are not enough $H^+$ ions in alkaline solutions, which enable a precipitation reaction according to Equations (9) or (10). Precipitation layers with a thickness between 10 μm and 20 μm are created by polarization at OCP + 4 V for 30 min. This layer thickness is approximately 1000 times greater than would be expected in the case of electrochemical passivation at the same anodic potential. In contrast to a dense and firmly adhering passive layer, the precipitation layers tend to be loosely adherent and porous and can easily be removed by rubbing. The precipitation layers are amorphous or nanocrystalline and mainly consist of the elements Al, O, and (in the case of phosphate-containing electrolytes) P. The Al:O ratio of about 2 to 3 measured by EDX and the results of the Raman measurements indicates that the layers formed after polarization in electrolyte 1 at OCP + 4 V probably consist of amorphous alumina or nanocrystalline γ-alumina. Sufficient agreement with the literature-reported Raman spectra of $Al(OH)_3$ c was not found. This contradicts the observations of Kurze that mainly $Al(OH)_3$ is formed due to anodic polarization in the potential range up to 75 V [13]. However, it cannot be ruled out that $Al(OH)_3$ or $AlO(OH)$ were present immediately after the polarization experiments and decomposed as a result of drying during sample storage. This is also indicated by the existence of a fine crack network, which probably arose as a result of the internal stresses caused by dehydration. The fine porosity may also have facilitated the drying of the layers and reduced the internal stresses so cracks were not widened significantly.

With increasing phosphate content in the electrolyte, the water-containing aluminum phosphate evansite $(Al_3(PO_4)(OH)_6 \cdot 6H_2O)$ is increasingly incorporated into the layer. This is in good agreement with Li et al., who describe the formation of alumina aluminum phosphate [25], with the difference that additional $OH^-$ ions and $H_2O$ molecules formed according to Equation (11) are also incorporated into the layer. It is likely that there was more intense drying in the course of the SEM investigations of Li et al. [25], which could have resulted in a more pronounced conversion of the evansite into alumina aluminum phosphate. In particular, a crack network, which can be seen in Figure 6, characterizes the layers that were produced in electrolytes 4 and 5 with a high phosphate content (right). Since the crack network is more pronounced compared to the layers of electrolytes with a

lower phosphate content, it can be assumed that the water-containing aluminum phosphate was at least partially dehydrated.

## 5. Conclusions

In addition to the existing state of knowledge, the formation of surface layers during polarization of a dual-phase steel in alkaline, aluminate-containing electrolytes in the potential range between OCP − 100 mV and OCP + 4000 mV was investigated. The potentiodynamic polarization was applicable to the screening of the potential range and potentiostatic polarization proved to be useful for the investigation of passivation or pH-induced precipitation at a distinct potential. The following potential ranges can be classified according to the dominating mechanisms:

1.  At an anodic potential of about 500 mV vs. SHE, slightly below the potential of oxygen evolution, electrochemical passivation takes place by the formation of an iron oxide, which probably consists of the maghemite phase.
2.  In the potential range between about 550 mV and 900 mV vs. SHE, passivation is still apparent. However, the passive layer is increasingly damaged with rising anodic potential due to oxygen evolution.
3.  At anodic potentials above about 1 V vs. SHE, oxygen evolution causes a sufficiently high pH drop at the anode surface, leading to the precipitation of a thick and porous oxide layer, which predominantly consists of amorphous alumina or nanocrystalline $\gamma$-alumina and, in the case of phosphate-containing electrolytes, the hydrous phosphate evansite.

According to the findings of this work, a pre-polarization step before the actual PEO process could be introduced in order to generate a precipitation layer with defined properties. For example, polarization at 4 V vs. SHE should be performed for at least 5 min in aluminate solution at pH 12 and for at least 15 min in an aluminate solution which additionally contains 0.1 mol/L phosphate at a similar pH. Furthermore, it was shown that this approach can also be applied to high-strength, multi-phase steels, even though the electrochemical behaviors of the ferrite and martensite phases differ significantly.

**Author Contributions:** Conceptualization, R.M.; methodology, R.M., C.A.R., F.S. and V.M.; validation, R.M., C.A.R., F.S., V.M., T.M. and T.L.; formal analysis, R.M., C.A.R., F.S. and V.M.; investigation, R.M., C.A.R., F.S. and V.M.; resources, T.L.; data curation, R.M., C.A.R., F.S. and V.M.; writing—original draft preparation, R.M.; writing—review and editing, R.M., C.A.R., F.S., V.M., T.M. and T.L.; visualization, R.M.; supervision, T.M. and T.L.; project administration, R.M, T.M. and T.L.; funding acquisition, T.L. All authors have read and agreed to the published version of the manuscript.

**Funding:** This research was funded by Deutsche Forschungsgemeinschaft (DFG, German Research Foundation), grant number 464291298.

**Data Availability Statement:** The authors confirm that the data supporting the findings of this study are available within the article.

**Acknowledgments:** The technical assistance of Anu George (polarization measurements, sample preparation, and optical microscopy), Saravanan Palaniyappan (Raman), and Marc Pügner (XRD) is gratefully acknowledged.

**Conflicts of Interest:** The authors declare no conflict of interest.

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
