# Peer review of "Passivation and pH-Induced Precipitation during Anodic Polarization of Steel in Aluminate Electrolytes as a Precondition for Plasma Electrolytic Oxidation"

_coatings, doi:10.3390/coatings13030656_

Round 1
Reviewer 1 Report
Dear sir
The results and its findings would be helpful for further processing of PEO. Some of the references can be improved and new citations one or two can be cited in this work. However, some typo errors like pH and its values., 500 mV vs. SHE. in conclusion part,
Missing the references No 18 in the last and first para of page 18 and 19.
significantly.
Author Response
Please find the detailed answer to your comments in the attachment.

Reviewer 2 Report
The information provided in connection with the coatings and corrosion is of great importance. But before the final decision, it needs many changes and corrections. After correcting the manuscript based on the comments, I will announce my opinion regarding the acceptance or rejection of the article. Consider all the comments below and highlight the changes.
1- The grammar and language structure of the article is inappropriate in some parts. It should be revised.
2- The title of the article is not appropriate. It is better to fix it.
3- Do not use long sentences in the abstract. The investigated parameters should be briefly stated in the abstract. In the abstract, state what parameters were used and what results were obtained. Extra sentences should be deleted.
4- The introduction is incomplete. In the introduction, the relationship between metallurgical parameters, microstructural changes and corrosion properties of steels has not been investigated. Introduction is not acceptable in its current format. Use the following articles to complete this section:
- https://doi.org/10.1111/ijac.13072 , - Study of the effect of temperature on corrosion behavior of galvanized steel in seawater environment by using potentiodynamic polarization and EIS methods, Materials Research Express, 6 (7), 2019, 076508. , - https://doi.org/10.1016/j.ijpvp.2022.104759
5- Specify the standard used to perform the electrochemical tests.
6- The analysis of polarization and OCP parts is incomplete. Use the following articles to complete this section:
- https://doi.org/10.1080/00325899.2021.1904582
- Study of the effect of temperature on corrosion behavior of galvanized steel in seawater environment by using potentiodynamic polarization and EIS methods, Materials Research Express, 6 (7), 2019, 076508.
Author Response

(The authors gave the same response as above.)

Reviewer 3 Report
In this paper, it explores Potentiodynamic and potentiostatic polarization tests in the potential range between open circuit potential (OCP) – 0.1 V and OCP +4 V were carried out to investigate the influence of the pH value and the phosphate content of aluminate-phosphate electrolytes on the formation of the surface layer simultaneously. This was done on a
high-strength dual-phase steel. Results show that at low potentials, iron hydroxides or oxides are initially formed. It was also shown that Precipitation leads to the formation of porous layers with thicknesses of 10 µm to 20 µm. The paper has some interesting results that could make it publishable in the journal of Coatings after the following major revisions:
1-Check the English of the whole paper.
2-Move these sentences from abstract to introduction:
“Studies show that unalloyed and low-alloyed steels “passivate” particularly well at the
beginning of the plasma electrolytic oxidation (PEO) in alkaline electrolytes containing aluminate ions. The underlying mechanism of the formation of the surface layer has not yet been finally clarified. Both a classic passivation by the formation of iron hydroxides or oxides and a precipitation reaction because of anodic oxygen development and the associated local pH reduction might already take place at anodic potentials of a few 100 millivolts to a few volts.”
2-Define in the abstract the range of the parameters that were varied, like ultrasonic power and frequency.
3-The last sentence of the abstract is not appropriate.
4-Introduction should be strengthened. The authors can use the following documents:
-(2022). Molecular insights into the adsorption of chloride ions in calcium silicate hydrate gels: The synergistic effect of calcium to silicon ratio and sulfate ion. Microporous and Mesoporous Materials, 345, 112248. doi: https://doi.org/10.1016/j.micromeso.2022.112248
-(2022). A Multifunctional Artificial Interphase with Fluorine-Doped Amorphous Carbon layer for Ultra-Stable Zn Anode. Advanced Functional Materials, 32(43), 2205600. doi: https://doi.org/10.1002/adfm.202205600
-(2022). A Functional Organic Zinc-Chelate Formation with Nanoscaled Granular Structure Enabling Long-Term and Dendrite-Free Zn Anodes. ACS nano. doi: https://doi.org/10.1021/acsnano.2c03398
5- figure 1 could be deleted. It’s a pretty known test.
6-formula #12, 13 could be transferred to the results section.
7-Consult the following references in the result and discussion section:
-(2023). Effect of aging plus cryogenic treatment on the machinability of 7075 aluminum alloy. Vacuum, 208, 111692. doi: https://doi.org/10.1016/j.vacuum.2022.111692
-(2023). Effect of heat treatment process on the micro machinability of 7075 aluminum alloy. Vacuum, 207, 111574. doi: https://doi.org/10.1016/j.vacuum.2022.111574
-(2020). Identification of the key host phases of Cr in fresh chromite ore processing residue (COPR). Science of The Total Environment, 703, 135075. doi: https://doi.org/10.1016/j.scitotenv.2019.135075
8-figure 4 does not have scale bar for all figures. Put scale bar if needed. Similarly put a scale bar for figures 6, 8, 10.
9-reference the formula that were used.
10-conclusion should be in bullet points.
Author Response

(The authors gave the same response as above.)

Reviewer 4 Report
I believe that Abstract and Introduction are presented very poorly and should be improved in accordance with the requirements for publications.
The novelty and actuality have to be emphasized especially clear. What is the difference between the results of this work and the data already known from literature?
1) I believe that Table 1 is needless. The composition of the steel could be shown in brackets, e.g., in line 134; “…pH-neutral NaCl solutions…” (line 497) better to say “in aqueous NaCl solutions (pH ~7)”.
2) NaAlO2 exist only in solid phase. This compound is well known to hydrolyzed in solutions to form hydroxy complexes of aluminum.
3) Figures 4 and 5 are of poor quality.
4) I don’t understand what do “-70±150 mV” and “-10±60 mV” (Table 3) signify?
5) Lines 497-506: In my opinion, it is better for these speculations to be somehow considered in the Introduction.
6) Lines 50-51: Here should be all the references supporting this statement.
Author Response

(The authors gave the same response as above.)

Round 2
Reviewer 2 Report
Although the authors have made changes to the article, the quality of this manuscript is very low. Also, the authors did not respond properly to my comments. The title of the article is not innovative. The introduction is incomplete. The analyzes of the corrosion section are incomplete. The correlation between microstructure and corrosion properties has been neglected. Overall, this manuscript is not suitable for the Coatings audience and is rejected in my opinion.
Author Response
Please find our detailed answer in the attachement.

Reviewer 3 Report
I have already stated my comments and they have to be taken into account during the revise, I believe.
In this paper, a review of three main recycling procedures for thermoplastic composites which are mechanical, thermal and chemical is presented. mechanical recycling of carbon fiber PEKK (CF/PEKK) thermoplastic composite is chosen to show the feasibility of recycling virgin and recycled materials are compared with dynamic mechanical analysis (DMA), differential scanning calorimetry (DSC), tensile and three-point bending test. The paper has some interesting results that could make it publishable in the journal of Coatings after the following major revisions:
1-The language of the paper in some parts require revision. Please check the English language of the paper.
2-The three sentences of the abstract should be deleted or shortened.
3-briefly mention the main findings in the abstract. Name the method employed to assess the various properties. Basically, define in the abstract what parameters were investigated, and what kind of tests were employed. The rest should be moved to introduction section.
4-Introduciton should be strengthened. To modify this section the following documents can be consulted:
-(2022). Experimental and DFT studies of flower-like Ni-doped Mo2C on carbon fiber paper: A highly efficient and robust HER electrocatalyst modulated by Ni(NO3)2 concentration. Journal of Advanced Ceramics, 11(8), 1294-1306. doi: 10.1007/s40145-022-0610-6
-(2023). Fresh, mechanical and microstructural properties of alkali-activated composites incorporating nanomaterials: A comprehensive review. Journal of Cleaner Production, 384, 135390. doi: https://doi.org/10.1016/j.jclepro.2022.135390
-(2022). Theoretical analysis on the lateral drift of precast concrete frame with replaceable artificial controllable plastic hinges. Journal of Building Engineering, 62, 105386. doi: https://doi.org/10.1016/j.jobe.2022.105386
5-it is should be “2. Recycling Procedures” .
6-Figure 3 could be deleted. It is a rather known process.
7-it should be “3.1 selection of materials”
8-figure 6 looks uncessary.
9-Consult the following documents in the discussion:
-(2022). Crushing and parametric studies of polygonal substructures based hierarchical cellular honeycombs with non-uniform wall thickness. Composite Structures, 299, 116087. doi: https://doi.org/10.1016/j.compstruct.2022.116087
-(2023). 3D fibrous aerogels from 1D polymer nanofibers for energy and environmental applications. Journal of Materials Chemistry A, 11(2), 512-547. doi: 10.1039/D2TA05984C
-(2022). An eco-friendly film of pH-responsive indicators for smart packaging. Journal of Food Engineering, 321, 110943. doi: https://doi.org/10.1016/j.jfoodeng.2022.110943
10-better define the rationale of the work in the last few sentences of the introduction.
11-when multiple figures are in one figure define them as “a”, “b”… and explain them in the caption, i.e. figures 7, 8.
12- put standard deviations in tables 1, and 2.
13-it should be “conclusions”.
14-make conclusions bullet points. The current version is long and tedious.
Author Response
We note that the review refers to another manuscript, which deals with the recycling of fibre-reinforced plastics. Probably, two different reviews have been mixed up during the uploading process. Please send us your comments regarding our manuscript on the passivation and pH-induced precipitation during anodic polarization of steel in aluminate electrolytes in the next round.
Reviewer 4 Report
OK.
Author Response
Thanks for taking your time and reviewing the revised version of our manuscript. We note that you still recommend "extensive editing of English language and style". Would you please tell us in more detail, which language and style issues should be fixed?
Round 3
Reviewer 3 Report
The paper can now be accepted for publication.
Author Response
Thank you for taking your time to support us with your remarks.